# Factors Affecting the Density of Corneal Endothelial Cells Cultured from Donor Corneas

**DOI:** 10.3390/ijms252211884

**Published:** 2024-11-05

**Authors:** Marina Bertolin, Alessandro Ruzza, Vanessa Barbaro, Elisa Zanetti, Diego Ponzin, Stefano Ferrari

**Affiliations:** Fondazione Banca degli Occhi del Veneto, Via Paccagnella 11, 30174 Venice, Italy; alessandro.ruzza@fbov.it (A.R.); vanessa.barbaro@fbov.it (V.B.); elisa.zanetti@fbov.it (E.Z.); diego.ponzin@fbov.it (D.P.); stefano.ferrari@fbov.it (S.F.)

**Keywords:** corneal endothelial cell (CEC) culture, endothelial cell density (ECD), endothelial-to-mesenchymal transition (ETM), cornea donors, transplantation

## Abstract

We investigated which specific correlation exists between the endothelial cell density (ECD) of corneal endothelial cell (CEC) cultures and the features of the donor corneas from which they originate. CEC cultures were prepared from one donor cornea or by pooling together cells of more corneas from elderly donors with ECDs lower or higher than 2000 cells/mm^2^. The ECDs of such primary cultures were evaluated and showed that that ECDs > 2000 cells/mm^2^ can be obtained only when CECs are isolated from (1) corneas of young donors; (2) at least two elderly donor corneas (if ECD > 2000 cells/mm^2^), or three elderly donor corneas (if ECD < 2000 cells/mm^2^). Secondary cultures are all characterized by ECDs < 2000 cells/mm^2^. Our study highlights the difficulties in obtaining cultures with ECDs > 2000 cells/mm^2^. Even if achievable with corneas from young donors, this becomes a challenging task when corneas from elderly donors are used (i.e., the overall majority of those collected by eye banks) and particularly when corneas from elderly donors with ECD < 2000 cells/mm^2^ are used. Pooling more of two corneas to obtain suitable CECs could technically overcome the problem. The above issues should be tackled appropriately before moving into clinical studies.

## 1. Introduction

Corneal transplantation is the treatment of choice for diseases caused by damage or dysfunction of corneal endothelial cells (CECs). Corneal endothelial cell density (ECD) decreases with age, from 2800–3000 cells/mm^2^ in healthy young people to 2200–2600 cells/mm^2^ in elderly people. It is assumed that 400–700 cells/mm^2^ is the density at which decompensation occurs [1]. Generally, endothelial cell dysfunctions are more evident in elderly people [2]. In addition to age, certain diseases or stress factors lead to a reduction in ECD [3,4,5,6]. Besides ECD, polymegathism (the coefficient of variation in cell area) and pleomorphism (the percentage of hexagonal cells) are morphological parameters commonly reported to indicate stress of the endothelium [7,8]. For eye banks, a healthy corneal endothelium with an ECD higher than 2000 cells/mm^2^ is a necessary condition to ensure the longest possible cornea graft survival after transplantation [9,10,11,12].

The shortage of donor corneas represents a relevant issue worldwide and, in the last decade, the regeneration of the corneal endothelium by means of cell therapy has been suggested as an alternative to endothelial keratoplasty. However, ideal and standardized CEC culture conditions have not been established yet, and no consolidated CEC-based therapy is yet available [13,14]. To date, only Kinoshita and colleagues have reported the results and follow-up studies of a clinical trial based on the injection of cultured human CECs obtained from young donors [15,16].

The most promising protocol for ex vivo cultures of CECs is based on the peel-and-digest method for cell isolation and the use of the dual media approach. The optimal seeding density for CECs obtained from young donors was shown to be no less than 10,000 cells/cm^2^, which allowed for regular passages in vitro and maintenance of cell specific morphological parameters [17,18,19,20,21,22,23,24]. Low densities, instead, were shown to accelerate CEC culture exhaustion [18]. CECs are known to have limited proliferative capacities and undergo rapid phenotype changes, a process known as endothelial-to-mesenchymal transition (EMT). As already reported, CEC cultures from young donors (<40 years of age) were shown to maintain both morphology and functionality for 3–4 passages in vitro [15,16,19,20,22,24,25,26,27,28,29,30,31,32,33], while those from donors > 40 years of age could only do so for 1–2 passages [34,35,36] before undergoing EMT. So far, no standard culture protocol has been established to overcome such limits.

While eye banks consider donor corneas as suitable for transplantation when ECDs are >2000 cells/mm^2^ [9,10,11,12], many of the studies published so far do not indicate the ECDs of the CEC cultures obtained. When reported (Table 1), they rarely exceed the value of 2000 cells/mm^2^. The parameters usually investigated in these papers are expression markers and morphological features (such as polymegathism and pleomorphism), while ECD has never been considered crucial to define the suitability of CECs. Following cornea transplantation, postoperative endothelial cell loss does occur, and clinical studies have shown that transplantation of endothelial grafts with ECDs lower than 2000 cells/mm^2^ is more likely to fail [9,10,11,12]. For these reasons, we think that the ECD of CECs should be carefully evaluated and considered as a crucial parameter to define the quality of the CEC cultures to be grafted. Following the indications of eye banks for donor corneas, we would suggest ECDs > 2000 cells/mm^2^ as the threshold value to define whether or not CEC cultures are suitable for transplantation.

Previous reports (detailed in Table 1) show that the ECD of CECs is correlated with quality standards such as morphology and expression markers. However, the presence of suitable morphological features and expression markers can also be found in CEC cultures with ECDs lower than 2000 cells/mm^2^. In a similar manner to what eye banks do when analyzing donor corneas, a threshold value for ECD should be identified, and only CEC cultures with ECDs > 2000 cells/mm^2^ should be deemed as suitable for transplantation, independent of the presence of regular morphological patterns and expression markers.

ECD of CECs has also been found to be correlated with other parameters such as donor age, ECD of the corneas used to isolate the CECs and the number of cells that have been plated [17,32,33,34,35,36,37]. However, to our knowledge, no studies have been reported linking such parameters to ECDs > 2000 cells/mm^2^ when CECs are isolated from donor corneas and primary cultures are obtained.

The aim of our study is therefore to evaluate whether corneas, which are unsuitable for transplantation (as having ECDs < 2000 cells/mm^2^), or corneas from elderly donors, which represent the overall majority of the tissues collected by eye banks, would allow the isolation of CECs, leading to in vitro cultures with ECDs > 2000 cells/mm^2^. The influence of parameters such as donor age and donor cornea ECD will also be evaluated (Figure 1). If such results were achieved, eye banks would have an important tool to increase the pool of corneas suitable for transplantation, either directly from deceased donors or as an advanced therapy medicinal product.

**Table 1 ijms-25-11884-t001:** Summary of studies reporting the characteristics of donor corneas and the endothelial cell density of corneal endothelial cell cultures.

References		Characteristics of Donor Corneas		ECD of CECs (Cells/mm^2^)	
Authors/Groups	Papers	Mean Age	Mean ECD (Cells/mm^2^)	P0	P2–P3
Mehta, J.S.	Frausto, R.F. 2020 [32]	2–35	>2300	1500 *	400 (P2) *
	Peh, G.S.L. 2015a [22]	2–37	2203–4425	217–451	154–426 (P3)
	Peh, G.S.L. 2015b [23]	3–39	2203–3968	730–1500	NA
	Peh, G.S.L. 2019 [30]	2–36	>2200	1100 *	NA
Kinoshita, S.	Kinoshita, S. 2018 [15]	7–29	NA	NA	1835–2530 (P2)
	Toda, M. 2017 [24]	14–58	2814–3969	2246 mean	NA
	Okumura, N. 2015 [26]	>40	NA	1100–2400	<500
Goldberg, J.L.	Bartakova, A. 2016 [19]	<50	2571–4425, for 2–34 donor age	1700–2400 *	NA
		preferred	1969–2865, for 38–77 donor age	(donor age not reported)	
Proulx, S.	Leclerc, V.B. 2018 [20]	20–71	NA	NA	400–500 * P2–P3
Melles, G.R.J.	Spinozzi, D. 2018 [35]	42–80	2000–2500 *	700–1000 *	600–1000 *
	Spinozzi, D. 2020 [38]	67 ± 12	NA	NA	1100–1300 *
Ferrari, S.	Parekh, M. 2019a [34]	65–78	1600–2100	1700–2400	500–1800
&	Parekh, M. 2019b [39]	71 ± 5	NA	2325–2631	NA
Ponzin, D.	Parekh, M. 2019c [40]	65–78	1600–2100	1850–2300	NA
Merra, A.	Merra, A. 2023 [41]	71 ± 5	1816	200 *	100–200 *
				2000 ≈ *	

Abbreviations. ECD: endothelial cell density; CEC: corneal endothelial cell; P0: first passage in culture; P2/P3: second and third passages in culture; NA: not applicable. * data inferred from the pictures/tables shown in the paper.

## 2. Results

### 2.1. CEC Cultures from the Cornea of a Young Donor Aged 5

ECDs of the CECs isolated from the control cornea were evaluated at each passage (P0 = 2400 ± 385 cell/mm^2^, P1 = 1830 ± 237 cell/mm^2^, P2 = 1502 ± 157 cell/mm^2^; P3 = 1296 ± 243 cell/mm^2^, P4 = 1333 ± 246 cell/mm^2^, P5 = 1005 ± 210 cell/mm^2^, P6 = 787 ± 228 cell/mm^2^), with EMT occurring at around the fourth passage.

### 2.2. Dual Medium Approach for CEC Cultures from Corneas of Elderly Donors

Compared to our standard protocol, the dual medium approach led to cultures characterized by significantly better parameters such as (1) percentage of polymorphic cells (34% ± 8 versus 47% ± 8, *p* = 0.02), (2) cell circularity (1.27 ± 0.03 versus 1.39 ± 0.09, *p* = 0.017) and (3) nucleus area (139 ± 26 µm^2^ versus 119 ± 16 µm^2^, *p* = 0.0001). No significant differences were seen in the mean cell area value and its coefficient of variation (standard deviation/mean cell area). The dual medium approach was therefore preferred to our standard protocol (i.e., endothelial growing medium throughout the culture, as previously described by Parekh 2019c) in all the experiments that we performed.

### 2.3. Plating Cell Density and ECDs of the Relative Cultures

Primary cultures with mean ECDs of 1247 ± 315, 1743 ± 284, 2163 ± 359 and 2512 ± 533 cells/mm^2^ were obtained from CECs plated at 500, 1000, 2000 and 4000 cells/mm^2^, respectively (Figure 2).

These results lead to a few important considerations. Firstly, we confirmed a direct correlation between the number of cells plated onto a well and the ECD of the primary cultures that were obtained. Secondly, we demonstrated that high plating densities are required to obtain primary CEC cultures with high ECDs. More specifically, we showed that a plating density of 2000 cells/mm^2^ is not enough to obtain primary cultures with ECDs > 2000 cells/mm^2^, and that one cornea from elderly donors with ECDs < 2000 cells/mm^2^ is unlikely to yield a primary CEC culture with ECDs > 2000 cells/mm^2^.

### 2.4. P0 CEC Culture with High ECD Pooling Cells from More Corneas

Old donor corneas with ECD < 2000 cells/mm^2^ (i.e., below the threshold set by eye banks for donor corneas to be used for transplantation) gave rise to CEC cultures having ECDs of 1476 ± 214, 1667 ± 324 and 3227 ± 773 cells/mm^2^ when 1:1, 2:1 and 3:1 plating ratios were used, respectively.

Old donor corneas with ECD > 2000 cells/mm^2^ (i.e., suitable for transplantation) gave rise to CEC cultures having ECDs of 2054 ± 371 and 2528 ± 376 cells/mm^2^ when 1:1 and 2:1 plating ratios were used, respectively. Results are shown in Figure 3 (P0 cultures, black columns) and compared to the ECDs of CECs isolated from one cornea of a 5-year-old donor, which was 2400 ± 237 cells/mm^2^.

CEC function was confirmed by ZO-1, Na+/K+-ATPase, Aquaporin and N-cadherin marker expression (Figure 4).

This series of experiments showed that primary (P0) CEC cultures with ECDs > 2000 cells/mm^2^ can be obtained only when:–CECs are isolated from corneas of young donors;–CECs are isolated and pooled together from at least two donor corneas from elderly donors (mean age 62.3 ± 9.2, 47–71 range), but only if the corneas have ECDs > 2000 cells/mm^2^;–CECs are isolated and pooled together from at least three corneas from elderly donors (mean age 72.6 ± 8.4, 52–79 range) with ECDs < 2000 cells/mm^2^ (bearing in mind that the latter can lead to several pieces of membrane being present in the cultures).

As corneas from young donors are less likely to be collected by eye banks, our results lead to important speculations. Our strategy based on the assumption that CEC cultures might eventually replace donor corneas (and help reduce the need for corneas for transplantation) will clearly not be viable when corneas from elderly donors with ECD > 2000 cells/mm^2^ are used, as more than one cornea would be needed.

In addition, even if CEC primary cultures can be obtained from three elderly donor corneas with ECD < 2000 cells/mm^2^ (three corneas from two donors) the issue of a potential immune response from more antigenic stimuli might arise.

### 2.5. CEC Cultures at P1 Passage

The following conditions and results were obtained (Figure 3, white columns):–Corneas from elderly donors, with ECD > 2000 cells/mm^2^ plated with a 2:1 ratio (i.e., two corneas to obtain one CEC culture) were able to generate primary and secondary cultures with ECDs of 2528 ± 376 cells/mm^2^ and 1213 ± 74 cells/mm^2^, respectively;–Corneas from elderly age donors with ECD < 2000 cells/mm^2^ plated with a 3:1 ratio (i.e., three corneas to obtain one CEC culture) were able to generate primary and secondary cultures with ECDs of 3227 ± 773 cells/mm^2^ and 1440 ± 127 cells/mm^2^, respectively. Membrane fragments were present in the primary cultures, but not in the secondary cultures.

Such data show that ECDs above 2000 cells/mm^2^ cannot be reached by secondary CEC cultures (P1) even when more corneas from elderly corneas with ECD > 2000 cells/mm^2^ are pooled together.

## 3. Discussion and Conclusions

The aim of our study was to correlate the ECD of donor corneas and the resulting in vitro CEC cultures. We mainly focused on elderly donor corneas with ECDs < 2000 cells/mm^2^ in order to evaluate the feasibility of a strategy aimed at reducing the demand for cornea transplantation through the supply of advanced therapy medicinal products made of CECs. However, to achieve such a goal, we have to take into consideration the following limitations: the endothelium area is around 1 cm^2^, donor corneas with ECDs < 2000 cells/mm^2^ have a total maximum number of 200,000 cells available to start a CEC culture and, finally, CEC cultures from elderly donors with low ECDs cannot go beyond the first passage, due to EMT. Such limitations eventually led us to the following conclusions: dealing with elderly donor corneas with low ECDs means working only with primary cultures and with a number of cells available to be plated that cannot exceed 200,000. Therefore, since this number of cells is not enough to obtain a CEC culture with ECD > 2000 cells/mm^2^, CECs will obviously have to be isolated from more than one donor cornea and pooled together.

Our results seem to confirm what was previously reported in the literature [22,32], i.e., that the ECD of CECs decreases when expanded in vitro. Our study demonstrates that the ECD of CEC cultures is lower than the ECDs of the donor corneas used to derive the cells from. Similar conclusions can also be extrapolated from other studies (see details in Table 1) that show how morphological characteristics and marker expressions decrease when the ECDs of the cultures decrease. However, they indirectly show that only cultures with very low ECDs (much lower than the 2000 cells/mm^2^ threshold we have set) do not have appropriate morphological characteristics and marker expressions, but also show that cultures with ECDs higher than 2000 cells/mm^2^ always have morphological characteristics and marker expressions of interest.

In addition to this, our results clearly show that cells from one single elderly donor cornea with low ECD are not enough to obtain a culture with ECDs higher than 2000 cells/mm^2^, thus supporting the rationale for higher cell plating densities. Considering that CEC cultures from old donor corneas can maintain their morphology and functionality only in the first passage, our results demonstrate that CEC cultures at P0 with high ECDs can be obtained only by pooling cells from multiple donor corneas. More specifically, our data clearly show that only specific conditions allow high ECDs to be obtained. As previously reported, primary corneal endothelial cell cultures with ECD > 2000 cells/mm^2^ can be obtained only if specific plating densities are used: if a young donor is used, one cornea is enough; if corneas from elderly donors with ECD > 2000 cells/mm^2^ are used, two corneas are needed; if corneas from elderly donors with ECD < 2000 cells/mm^2^ are used, up to three corneas are needed.

The following conclusions can be drawn from our data.

Corneas from young donors are unlikely to be regularly obtained by eye banks. The average age of the donors in our eye bank in Venice, Italy, is 64.7 ± 12.3 (14,095 donors over the years 2017–2021). When corneas from young donors are available, they usually have ECDs much higher than 2000 cells/mm^2^ and are therefore used for transplantation. While potentially interesting, the strategy of relying on corneas from young donors is therefore unlikely to be implemented, at least in many European eye banks. Using corneas obtained from elderly donors would instead allow a regular and constant reservoir of CECs to be available to obtain cultures. Not surprisingly, many groups prefer working with corneas from young donors (REFS Kinoshita, Metha and Golberg), and when those from older donors are used, corneas with high ECDs (>2500 cells/mm^2^) are preferred (REFS Proulx and Melles) (Table 1). To our knowledge, only Parekh and colleagues (REF) systematically investigated the use of corneas from donors aged 50 to 80 years old with ECDs < 2000 cells/mm^2^ as a way for eye banks to save precious tissues that would otherwise be discarded [34,36,40,41,42]. Moreover, human CEC cultures obtained from elderly donors generally show greater heterogeneity than those from younger donors [17].

If donor corneas with ECD < 2000 cells/mm^2^ (i.e., unsuitable for transplantation) were used to isolate CECs, such a strategy would allow precious donated tissues to be saved that would otherwise be discarded. Unfortunately, our results show that CECs would have to be pooled from at least three tissues. This opens up the question of whether such a strategy would increase the risk of rejection/failure following transplantation, because of a higher immune response towards antigenic stimuli from more donors. To our knowledge, there are no consensus papers on the immunologic complications of CECs and cornea graft transplantations, particularly when multi-donor transplantations are carried out [43,44,45]. If such studies should move toward a clinical application, such a strategy will have to be investigated in depth.

As detailed elsewhere, cultures of CECs might be useful for reducing the shortage of donor corneas. However, if two corneas obtained from elderly donors with ECD > 2000 cells/mm^2^ (therefore suitable for transplantation onto two recipients) are needed to obtain one primary CEC culture with the same ECD, the advantages of this strategy become irrelevant as it is not able to support the need for corneal tissues.

A further conclusion is that, once plated, CECs obtained from elderly donor corneas hardly proliferate, but rather reorganize. Our results do not come as a surprise, as similar conclusions were also drawn when preclinical tests for Descemet’s Stripping Only (DSO) and Descemetorhexis without Endothelial Keratoplasty (DWEK) were carried out. Cell migration tests performed in ex vivo [38,46,47,48,49] or in vivo [50,51,52] settings showed lower ECDs of the restored endothelium. Our results seem to confirm the hypothesis that when CECs form a layer, they prefer to expand rather than proliferate. Our study once again shows that the ideal culture conditions have not been set yet, thus suggesting that the mechanisms underlying CEC physiology remain to be elucidated.

## 4. Materials and Methods

### 4.1. Characteristics of Donor Corneas

Donor corneas were collected by Fondazione Banca degli Occhi del Veneto (Venice, Italy) and used for the validation studies described in the manuscript, with written informed consent from the donor’s next of kin. Post-mortem time (i.e., time from explant to isolation of cells) was less than 24 h. A total of *n* = 95 corneas from 61 different elderly donors (mean age 67.6 ± 9.2 years—ranging from 47 to 79 years) were used, with ECDs that were considered high (>2000 cell/mm^2^) or low (<2000 cell/mm^2^). As a control, we used a cornea from a 5-year-old donor (ECD = 2800 cell/mm^2^). CEC cultures were established within 1 month from the donor date of death. All the donor corneas used in the study complied with the quality standards required by the guidelines for donor cornea transplantation set by the Competent Authorities, including the compulsory safety analyses (serology, microbiology and virology) required for inclusion of the donors.

### 4.2. Corneal Endothelial Cell Cultures

Our standard CEC culturing protocol [39] corresponds to that described by Parekh and colleagues [34] and was here compared to a revised dual medium approach described by Peh et al. [22,23]. Briefly, the Descemet membrane was stripped and then incubated with 2 mg/mL type 1 collagenase in 15 mL of human endothelial SFM medium (Thermo Fisher Scientific, Waltham, MA, USA) for 2–3 h at 37 °C followed by 1.5 mL of 1× TrypLE Express for 5 min at 37 °C. The isolated cells were centrifuged for 5 min at 800 rpm and pellets re-suspended with our standard endothelial growing medium (1:1 HamF12 and M199 (Sigma-Aldrich, St. Louis, MO, USA)) containing 5% FBS (Sigma-Aldrich, St. Louis, MO, USA), 20 µg/mL of ascorbic acid (Sigma-Aldrich, St. Louis, MO, USA), 1% insulin–transferrin–selenium 100× (Thermofisher Scientific, Waltham, MA, USA), 10 ng/mL of FGF basic (Thermofisher Scientific, Waltham, MA, USA), 10 µM of ROCK inhibitor (Y-27632; Miltenyi Biotec, Bergisch Gladbach, Germany) and 50 μg/mL of penicillin–streptomycin (Euroclone, Milan, Italy) and then plated (i.e., primary culture or P0) onto a precoated surface (FNC coating MIX, US Biological, DBA, Milan, Italy).

Considering that the area of a regular human cornea is approximately 1 cm^2^, primary CECs (P0) were plated at a 1:1, 2:1 or 3:1 ratio, i.e., all CECs from 1, 2 or 3 donor corneas, respectively, were plated onto a 1 cm^2^ plate surface (that corresponds to the corneal posterior surface area). CECs were plated in 0.35 mm^2^ wells of the 4-well IBIDI plate (Culture-Insert 4-well in µ-Dish 35 mm, IBIDI, Gräfelfing, Germany): 1:1, 2:1 or 3:1 ratio pools of cells were equally split onto three 0.35 mm^2^ IBIDI wells. As detailed earlier, three 0.35 mm^2^ IBIDI wells correspond to approximately 1 cm^2^, just like the area of the corneal posterior surface.

The medium was changed every other day. When subconfluent (90%), CECs at P0 (i.e., primary culture) were cultured for further 24–48 h before undergoing immunohistochemistry analyses or being passaged (plating of P1 secondary cultures). In order to reduce any potential immune response, the CEC pool from 2 donor corneas refers to the isolation of cells from 2 corneas of the same donor (1 antigenic stimulus), while the CEC pool from 3 donor corneas refers to the isolation of cells from 2 donors (2 possible antigenic stimuli).

### 4.3. Dual Medium Approach

CECs grown and maintained through a revised dual medium approach [22,23] were firstly plated in the presence of our standard endothelial growing medium and, once subconfluent (90%), with the endothelial basic medium, i.e., 1:1 HamF12 and M199 (Sigma-Aldrich, St. Louis, MO, USA) containing 5% FBS (Sigma-Aldrich, St. Louis, MO, USA) and 50 μg/mL penicillin–streptomycin (Euroclone, Milan, Italy). The dual medium approach was used for all the experiments described, while the standard protocol was used in the first set of experiments (comparison of the two culturing conditions).

### 4.4. Plating Cell Density and ECDs of the Relative Cultures

We investigated which specific correlation exists between the number of cells plated and the ECDs of the cultures that originate. Such information is needed to evaluate the lowest plating density required to obtain primary cultures with ECDs > 2000 cells/mm^2^ when using corneas from elderly donors and with ECD < 2000 cells/mm^2^ as the starting material. Therefore, CECs isolated and pooled from corneas obtained from elderly donors were plated at the following densities: 500, 1000, 2000 and 4000 cells/mm^2^ (*n* = 5 for each plate density condition).

### 4.5. P0 CEC Cultures with High ECD

We tried to define the number of corneas to pool together in order to obtain primary CEC cultures with the desired ECD (i.e., >2000 cells/mm^2^). We compared 5 conditions (each of them *n* = 6): corneas from elderly donors both with ECD < 2000 cells/mm^2^ (i.e., below the threshold set by eye banks for transplantation) and ECD > 2000 cells/mm^2^ (i.e., ECD suitable for transplantation) plated at 1:1 and 2:1 ratios (i.e., all CECs from 1 and 2 corneas, considering a total plating surface of 1 cm^2^). For corneas from elderly donors with ECD < 2000 cells/mm^2^, we also tested the 3:1 ratio.

### 4.6. Secondary CEC Cultures (P1)

To evaluate the characteristics of secondary CEC cultures (P1), we only used primary cultures with ECDs > 2000 cells/mm^2^. This includes P0 CEC cultures obtained from corneas with ECD ≥ 2000 cells/mm^2^ with a 2:1 plating ratio and P0 CEC cultures obtained from corneas with ECD ≤ 2000 cells/mm^2^ with a 3:1 plating ratio. We examined *n* = 3 P1 cultures for both conditions. P0 to P1 passages were performed by treating each P0 culture with 200 µL of 1× TrypLE Express for 5–8 min at 37 °C [39]. Cells were split at a 1:2 ratio to obtain secondary cultures.

### 4.7. Immunofluorescence

Human CEC cultures were stained for ZO-1 (Alexafluor488-conjugated primary antibody, Thermo Fisher Scientific, Waltham, MA, USA), Na+/K+ -ATPase (Santa Cruz, CA, USA), Aquaporin 1 (Abcam, Cambridge, UK) and N-cadherin (Abcam, Cambridge, UK). Briefly, the samples were fixed with 1% paraformaldehyde (PFA, Santa Cruz, TX, USA) for 20 min, permeabilized with 0.5% Triton (Sigma-Aldrich, St. Louis, MO, USA) in 1× PBS for 7 min, blocked with 10% goat serum (Sigma-Aldrich, St. Louis, MO, USA) in 1× PBS for 1 h at room temperature (RT), incubated with primary antibodies in 10% goat serum (anti-ZO-1: 1:200 for 3 h at RT; anti-Na+/K+-ATPase: 1:50 for 1 h at +37 °C; anti-Aquaporin and anti-N-cadherin: 1:100 for 1 h at +37 °C). Secondary antibodies were then incubated for 1 h at room temperature (goat anti-rabbit and goat anti-mouse were both obtained from Invitrogen, San Diego, CA, USA) and finally mounted with a medium containing DAPI (EMS, Società Italiana Chimici, Rome, Italy). Immunofluorescence was evaluated by means of an Eclipse Ti Nikon microscope (Nikon, Amstelveen, the Netherlands). The Eclipse Ti Nikon microscope program was used to obtain measurements. ZO-1 stainings were used to investigate parameters such as the cell area, ECD, circularity, polymorphism and nuclei area; three fields were analyzed for each sample. ZO-1, Na+/K+-ATPase, Aquaporin and N-cadherin stainings were used to confirm CEC culture functions.

### 4.8. Statistical Analysis

Results are expressed as mean ± SEM. The non-parametric Wilcoxon and Kruskal–Wallis tests were used to compare differences between groups. The level of significance (*p*) was set at <0.05.

## Figures and Tables

**Figure 1 ijms-25-11884-f001:**
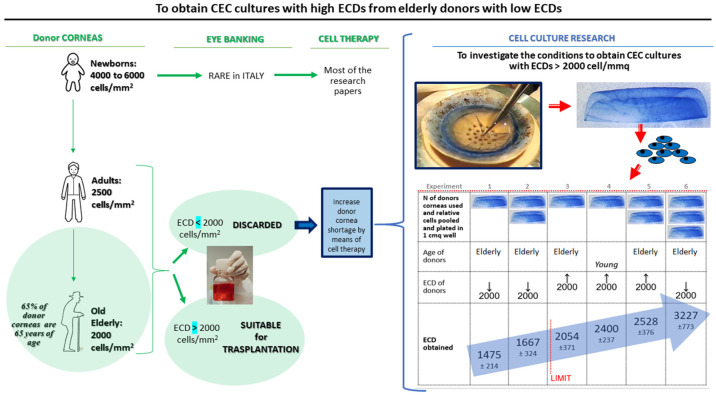
Schematic representation of the research idea, objective, experiments and results.

**Figure 2 ijms-25-11884-f002:**
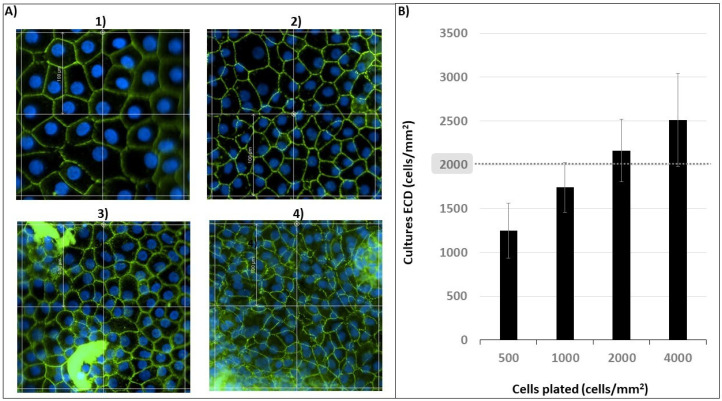
Number of cells (CECs) plated and the endothelial cell density (ECD) of the cultures obtained. (**A**) Photos of the cultures. (**B**) Graphical representation of the data. CECs were plated at 500 (**A1**), 1000 (**A2**), 2000 (**A3**) and 4000 (**A4**) cells/mm^2^; relative ECDs obtained: 1247 ± 315, 1743 ± 284, 2163 ± 359 and 2512 ± 533 cells/mm^2^. Magnification: 20×; scale bar: 100 µm.

**Figure 3 ijms-25-11884-f003:**
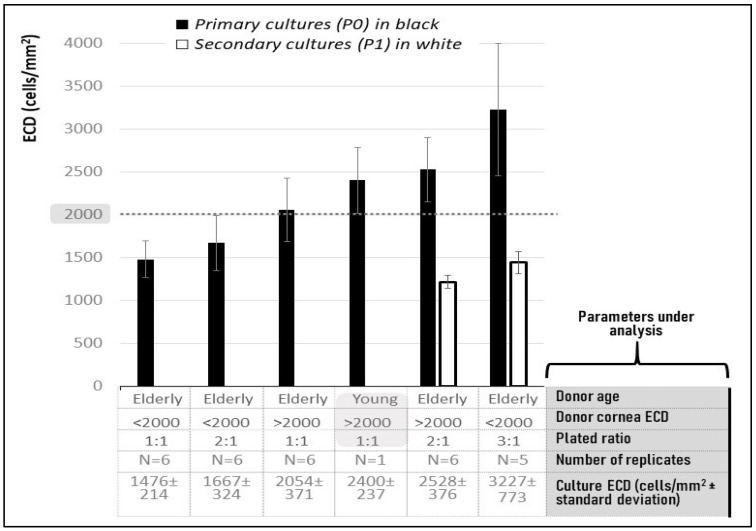
Endothelial cell density (ECD) of corneal endothelial cell (CEC) cultures. Primary (P0) and secondary ECD cultures (P1) obtained investigating different donor corneas parameters. 1:1, 2:1 and 3:1 plating ratios were investigated too, i.e., all CECs from 1, 2 or 3 donor corneas in a well of 1 cm^2^ plating area (corresponding approximately to the area of a regular human cornea).

**Figure 4 ijms-25-11884-f004:**
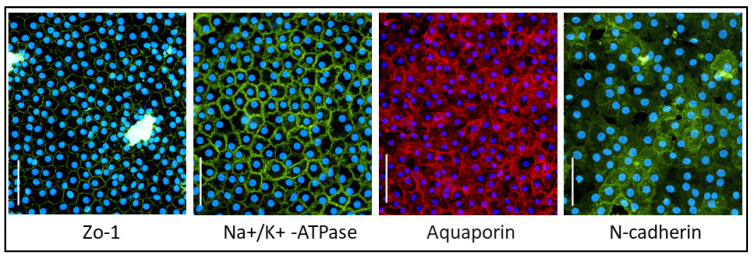
Marker expression. Representative primary (P0) CEC cultures and marker expression for ZO-1, Na+/K+-ATPase, Aquaporin and N-cadherin. Scale bar: 100 µm.

## Data Availability

The raw data supporting the conclusions of this article will be made available by the authors on request.

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
