# Peer review of "Factors Affecting the Density of Corneal Endothelial Cells Cultured from Donor Corneas"

_ijms, 2024, doi:10.3390/ijms252211884_

Round 1

Reviewer 1 Report (Previous Reviewer 1)

Comments and Suggestions for Authors

Thank you for submitting the article “Factors Affecting The Density Of Corneal Endothelial Cells Cultured From Donor Corneas” to MDPI Molecular Sciences.

The article discusses if corneas unsuitable for transplantation would allow the isolation of CECs leading to in vitro cultures with ECDs > 2000 cells/mm2 for eye banks to increase the pool of healthy cornea. The study is relevant and significant because it details procedures for endothelial cell cultures which may avoid wasting of donor corneas as a consequence of inappropriate culture protocols.

The study has certain limitations which should be clearly addressed:

-          Verification and study of a larger number of cases are required to demonstrate the reliability of the results. To formulate an open question: What is needed exactly to confirm the findings of your study and legitimate a change in guidelines?

-          Further considerations should be given to the pooling of cells: Could pooling lead to immunologic phenomena and how should this be excluded experimentally? In addition, the medico-legal side has to be addressed due to the fact that pooling could lead to an increased statistical risk of transmission of infected cells.

-          What are the implications of the results of the study on ethics for cell culture and transplantation? Is a revision of current guidelines required?

Concerning the style: Please remove phrases which appear to express an opinion (“we believe…”).

Please complete the publisher statements at the end and remove sample text (especially funding and ethics)!

Author Response

Reviewer 1

1.1)

Verification and study of a larger number of cases are required to demonstrate the reliability of the results.

We agree with the reviewer that a larger number of cases will made the results more confident. It has moreover to be considered that these experiments require human donor corneas and their availability is limited. According to the Italian Law 91/99, human donor corneas are collected exclusively for transplantation. Tissues that cannot be used for transplantation due to biological parameters (i.e., low endothelial cell densities, stromal scarring, etc.) or anamnesis (i.e., unsuitability of donor characteristics to transplantation) can be used for research purposes if the aims of the study are to improve transplantation. In addition, the raise of lamellar keratoplasty techniques as opposed to the transplantation of a whole cornea (penetrating keratoplasty) is becoming a way to use corneas that in the past would have never been used, as the layers altered are discarded while the suitable ones are used (for example, corneas with low endothelial cell densities can be used for Deep Anterior Lamellar Keratoplasty).

This clearly limits the number of tissues that can be used for studies like ours to very little.

To our knowledge, such issue is a limitation to all the studies similar to ours, as examples detailed below:

Peh and colleague:

  • in ref. 22 (https://journals.sagepub.com/doi/full/10.3727/096368913X675719 ) Table 1 shows that they used a total of 21 corneas and most of the experiments were n=3/5;
  • in ref. 23 (https://www.ncbi.nlm.nih.gov/pmc/articles/PMC4387913/pdf/srep09167.pdf ), Table 2, 3 and 4 show that part of the experiments performed were n=4 and n=5:

Spinozzi and colleague:

  • 50 used n=5 (https://link.springer.com/article/10.1007/s10561-020-09854-z ),
  • 35 used n=7 (https://pubmed.ncbi.nlm.nih.gov/29043524/)
  • in paper https://www.ncbi.nlm.nih.gov/pmc/articles/PMC7586272/ they used a total of 12 cornea and Table 2 reports n=5, n=4 and n=3 for the different experiment reported;

As detailed in the Materials and Methods we used a total of n=95 corneas from 61 different donors and the experiments were carried out with n=5 (2.4 Plating cell density and ECDs of the relative cultures), n=6 (2.5 P0 CEC cultures with high ECD); and test 2.6 Secondary CEC cultures (P1) was n=3 but this condition was immediately discarded.

It is to be appreciated that we also used a paediatric cornea, which is an exceptional event as whenever this happens (very rarely) a call to all the hospitals in Italy is made in order to make such corneal tissue available to the largest population of patients ever.

In conclusion, while it would be desirable to use more corneas to corroborate our results, it would be NOT ETICAL to use corneas that can be used for transplantation. We feel that the numbers used in our study exceeds the numbers used by other authors in previous studies.

To formulate an open question: What is needed exactly to confirm the findings of your study and legitimate a change in guidelines?

There are no guidelines detailing the ECD of the cells that are injected or transplanted in cell therapy approaches and our intention is not to define a consensus on a field that it is still in its infancy.

As follow-ups of clinical studies with donor corneas clearly demonstrate a drop in ECDs after tissues are transplanted, we sought to demonstrate that the use of conditions allowing to transplant grafts of cultured CECs with ECDs> 2000 cells/mm2 might be beneficial in the long run.

1.2)

Further considerations should be given to the pooling of cells:

Could pooling lead to immunologic phenomena and how should this be excluded experimentally?

Patients who have had transplantation of donor corneas in both eyes are likely to receive the tissues from two different donors and, to our knowledge, there is no an increased risk of rejection.

In the discussion, we mentioned the possibility that pooling cells from corneas of different donors might lead to increased risks of immune responses, but whether this is to be considered as a consequence would involve lengthy immunology-based experiments on animal models that we cannot foresee at the moment and that are beyond the scopes of this paper.

In addition, the medico-legal side has to be addressed due to the fact that pooling could lead to an increased statistical risk of transmission of infected cells.

Cells are isolated from donor corneas collected following the rigid donor selection procedures of the eye bank. As corneas are collected solely with the purpose of transplantation, lack of microbiological and virological contaminations are compulsory conditions. Therefore, isolating cells potentially leading to transplantation of infected cells and transmission of disease has the same risk of cornea transplantation. This point was discussed at lines 103-5. We reviewed this in order to make this message more clear.

1.3)

What are the implications of the results of the study on ethics for cell culture and transplantation?

There are no ethical implications similarly to what is currently done with transplantation of corneal epithelial cells for limbal stem cell deficiency. The aim of our paper is to identify a quality control parameter (ECD) that might lead to better follow up results following transplantation, similarly to what observed when donor corneas are grafted.

Is a revision of current guidelines required?

There is no need for a revision because so far there are not current guidelines regarding the ECDs of CEC cell therapy-based approaches. The only protocol approved is that developed by Kinoshita and colleagues, but they do not graft in vitro regenerated CEC layers, but inject CECs from young donors and no limits are set for the ECD of the CECs used for injection.

1.4)

Concerning the style: Please remove phrases which appear to express an opinion (“we believe…”).

We agree with the reviewer and have reviewed the expressions “we believe…” and similar ones (lines 65, 67 71-5) in the text.

Reviewer 2 Report (Previous Reviewer 2)

Comments and Suggestions for Authors

The authors point out that ECD >2000 cells/mm2 can only be obtained when CECs are isolated from young donor corneas, which is difficult with CEC isolation from older donor corneas, and suggest measures such as pooling two corneas.

It is understood that more minor points are addressed than in previous manuscripts. However, although the authors' culture techniques showed results for cell density and ZO-1, one of the cell-to-cell binding factors, the lack of additional experiments on other CECs reported in the manuscript, such as N-cadherin, Na+/K+/ATPase, and SLC4A11, which are involved in CEC function, is disappointing. Again, no additional experiments have been performed on the authors' culture methods. Once again, I recommend that additional experiments be performed to demonstrate that the authors were able to confirm the expression of these markers in CECs that became confluent with their culture methods.

Author Response

Reviewer 2

Although the authors' culture techniques showed results for cell density and ZO-1, one of the cell-to-cell binding factors, the lack of additional experiments on other CECs reported in the manuscript, such as N-cadherin, Na+/K+/ATPase, and SLC4A11, which are involved in CEC function, is disappointing. Again, no additional experiments have been performed on the authors' culture methods. Once again, I recommend that additional experiments be performed to demonstrate that the authors were able to confirm the expression of these markers in CECs that became confluent with their culture methods.

WE apologize with the reviewer for misunderstanding the requests made in the previous revisions. We of course did perform such experiments and tested such markers in our cultures. We have now added an extra figure (Figure 4) with Zo-1, N-cadherin, Na+/K+ -ATPase and Aquaporin expressions (lines 227-228 and 235-237). MATERIALS and METHODS 2.7 Immunofluorescence section was reviewed as well (lines 165-180).

Round 2

Reviewer 2 Report (Previous Reviewer 2)

Comments and Suggestions for Authors

The authors take the reviewers' suggestions for revisions seriously. This paper will be ACCEPTED.

This manuscript is a resubmission of an earlier submission. The following is a list of the peer review reports and author responses from that submission.

Round 1

Reviewer 1 Report

Comments and Suggestions for Authors

Thank you for submitting the article “Factors Affecting The Density Of Corneal Endothelial Cells Cultured From Donor Corneas” to MDPI International Journal of Molecular Sciences. Please address the following comments and questions:

The introduction should be extended, specifically describing the rationale and need for the project. What is known about corneal transplants? What are the problems, ethical issues, regulations and laws on eye banking and corneal processing? Please provide a clear hypothesis and research questions. The aim of the study should better be mentioned here (instead of discussion section).

Please introduce/define the terms “primary” and “secondary culture”.

What is the need and what is the ethical justification for pooling the endothelial cells derived from the corneas? Is this procedure in line with current standards of good practice? Can you confirm that through your method, corneas are not wasted?

What is behind the “magical number” of 2000 cells/mm2? Where does this cut-off come from? what is its use? Please provide further explanation and references. Is this really important or could you also transplant a cornea with less cells? Would it not be better to, for instance, age-match corneas rather than attempting augmentation of the cell numbers (a process by which more corneas are used)?

Figure 1: Caption should be changed to a precise and concise description of the figure (“graphical abstract. Schematic representation of the paper.” is not very informative).

What are common complications of the approach (please mention all relevant ones). What is the risk of bacterial contamination according to studies? Are there immunologic complication? Are the complications expected to be higher in pooled cultures compared to single-cornea cultures? (Please provide relevant references).

Is the donor cell density solely a function of age or are other parameters relevant? Can you provide references?

Contamination issue in pooled transplants: How would you trace an infection such as HIV (although unlikely) if multiple sources for cell-culture are used? How could regulators be convinced that this is not an issue?

More information on pooled cells is required. Is it possible that pooled cultures could behave differently than single-source cultures? Please describe challenges, risks and experiments required (including relevant references from the literature) to anticipate their behaviour. What did the experiments show in terms of quality of the cells (for instant polymegathism and pleomorphism) which are common in elderly corneas?

Please explain further why corneas are pooled for cell culture. Why is it not possible to expand to >2000 cells/mm2 from one cornea with <2000 cells/mm2? What are the limitations? (This should maybe be explained in the introduction section because it is a lab-specific starting point for the research.) How do the authors consider the option of finding a way to expand cells further instead of pooling them (please reference currently available literature and provide well-founded interpretations and assumptions).

Please evaluate and assess risk and relevance for presence of membranes in pooled transplants (especially from a surgeons/clinician’s perspective)?

L90-94: this paragraph is slightly unlogic. What is the exact argument that CEC cultures will not replace corneal transplants?

L150-152: How is the conclusion (minimum 2000 cells/mm2) derived exactly (“Therefore, following the steps of eye banks, we should consider an ECD of > 2000 cells/mm2 as the threshold value to define whether CEC cultures are suitable or not for transplantation.”)

Table 1: What do the columns “Author” and “Reference” mean?

Avoid too many bullet point listings (L83-88 and 165-170 is very similar).

L170: avoid too prominent rhetorical questions, this sounds overly scholastic.

Can the conclusion of the research be simplified to “the more cells I put into the culture, the higher the yield (output or number of cells in the end)”? Is this conclusion a novelty (given the fact that it is known that endothelial cells die in culture especially if they are from an older donor)? If this is not a novelty, what is the novelty of the study? Is the “operational” cut-off of 2000cells/mm2 maybe unimportant in this case?

Structure of the article: Traditionally, the Materials and Methods section comes at second position after the introduction. I would advise to move the Material and Methods section to the second position as understanding the methodological approach is important for comprehension of the results section which should follow after.

Please complete the publisher statements at the end and remove sample text!

Comments on the Quality of English Language

English is generally fine, minor corrections needed. 

Reviewer 2 Report

Comments and Suggestions for Authors

A reviewer can understand what the authors want to report, but the experimental methods and what they prove are too few. Other similar studies have been reported (Merra A et al. Exp Eye Res 2024, doi: 10.1016/j.exer.2024.109815.) that validate many items, and the novelty is unclear.

Unfortunately, this manuscript only demonstrates cell number and proof of their proof as corneal endothelial cells by expression of ZO-1. In other studies, corneal endothelial cells are usually verified by N-cadherin, which proves intercellular junctions, Na+ K+/ATPase and SLC4A11, which are involved in function, in addition to ZO-1. Conversely, EMT markers (such as α-SMA and CD44) altered from corneal endothelial cells must also be demonstrated.

It has also been reported that the addition of ROCK inhibitors does not favor cultures established from older donor corneas, as Y-27632 has been shown to increase HCEnC proliferation and adhesion only in cultures derived from young donors. (Peh GS et al. Sci Rep 2015, doi: 10.1038/srep09167.) In cell culture experiments, it is very important to describe the coating conditions of the dish.